# How Far Are We from True Unlearnability?

**Kai Ye[†], Liangcai Su[†], Chenxiong Qian[✉]**
The University of Hong Kong

## Abstract

High-quality data plays an indispensable role in the era of large models, but the use of unauthorized data for model training greatly damages the interests of data owners. To overcome this threat, several unlearnable methods have been proposed, which generate unlearnable examples (UEs) by compromising the training availability of data. Clearly, due to unknown training purposes and the powerful representation learning capabilities of existing models, these data are expected to be unlearnable for models across multiple tasks, i.e., they will not help improve the model's performance. However, unexpectedly, we find that on the multi-task dataset Taskonomy, UEs still perform well in tasks such as semantic segmentation, failing to exhibit *cross-task unlearnability*. This phenomenon leads us to question: *How far are we from attaining truly unlearnable examples?* We attempt to answer this question from the perspective of model optimization. To this end, we observe the difference in the convergence process between clean and poisoned models using a simple model architecture. Subsequently, from the loss landscape we find that only a part of the critical parameter optimization paths show significant differences, implying a close relationship between the loss landscape and unlearnability. Consequently, we employ the loss landscape to explain the underlying reasons for UEs and propose Sharpness-Aware Learnability (SAL) to quantify the unlearnability of parameters based on this explanation. Furthermore, we propose an Unlearnable Distance (UD) to measure the unlearnability of data based on the SAL distribution of parameters in clean and poisoned models. Finally, we conduct benchmark tests on mainstream unlearnable methods using the proposed UD, aiming to promote community awareness of the capability boundaries of existing unlearnable methods.

## 1 Introduction

High-quality data is the new oil of our era, giving rise to a series of impressive works such as large language models (LLMs). In pursuit of the Scaling Law, publicly available but unauthorized data may be unwittingly used during model training, severely harming the interests of data owners. To avoid this threat, Unlearnable Examples (UEs) have been proposed, which actively disrupt data usability in training by adding imperceptible perturbations. Existing unlearnable methods have deeply explored images, rendering unlearnable examples almost unhelpful for training image classification models and exhibiting *single-task unlearnability*. However, once data is public or leaked, its training purpose is **unknown**, meaning the data may be used to train various models with different tasks, e.g., semantic segmentation and object detection. However, *multi-task unlearnability* of UEs has not only been understudied but also largely overlooked.

Furthermore, we investigate the multi-task unlearnability of unlearnable examples, generated by unlearnable methods based on proxy models and heuristics. For the former, constructing UEs necessitates model training to identify effective perturbations, such as EM (Huang et al., 2021) In contrast, heuristic methods introduce specific bias information, assisting the model in finding "shortcuts" during training and disregarding the data's inherent features, such as OPS (Wu et al., 2022). Since heuristic methods primarily cater to classification tasks, we employ the representative unlearnable methods (EM, OPS and AR) to generate UEs and evaluate the performance gap between its poisoned model and clean model under multi-task scenarios, as depicted in Figure 6 in Appendix A.1.

---

[†] Contributed Equally. [✉] Corresponding Author (`cqian@cs.hku.hk`).

We also compare the performance of different tasks during EM (Error-Minizing) training, including similar tasks and tasks with significant differences. The results show that the EM still fails to exhibit unlearnability consistent with the nature of the tasks under multi-task scenarios, as shown in Figure 7 in Appendix A.1. Surprisingly, on the Taskonomy dataset (Zamir et al., 2018), the UEs generated by existing methods have a minimal negative impact on model performance, suggesting that existing UEs do not cause model training to fail. In other words, current Unlearnable Examples have not genuinely achieved multi-task unlearnability. Furthermore, the experimental results also demonstrate that existing UEs fail in cross-task scenarios (see Appendix A.2). This observation raises the question: *How far are we from attaining truly unlearnable examples*?

To answer this question, we need a reasonable metric to evaluate the unlearnability of UEs. Existing unlearnability evaluation depends on the performance of models after training, for example, the difference in accuracy between models trained on UEs and clean samples in classification tasks. However, this metric cannot explain the cause of UEs and relies on the specific form of downstream tasks. Furthermore, we observed that the model parameters trained on clean datasets have higher values than those trained on UEs. This implies that with the same model initialization, the parameter updates of the poisoned models are slower and the magnitude of updates is lower, as shown in Figure 9 in Appendix A.3. Therefore, unlike existing work, considering that UEs directly impact the parameter update process during model training, we attempt to assess data unlearnability from the perspective of model optimization.

As a result, we explore the connection between unlearnability and the training process. First, we observe the model convergence process on clean samples and UEs through the loss landscape. We find that only a small number of key parameters exhibit significant differences between the two models, and these key parameters cannot converge on UEs. This suggests that there is a more direct connection between parameter updates and unlearnability. We then attempt to describe this connection, stating that if a sample is unlearnable, the direction of parameter updates should move along contour lines, or the magnitude of parameter updates should be so small that they are almost zero, meaning the loss can be considered as not decreasing. In practice, it is difficult for parameters to update strictly along contour lines, but the latter may be feasible under approximate conditions. Specifically, if the parameters are in a globally flat area with sparse contour line density, the impact of parameter updates on the loss is relatively negligible.

Building on this understanding, we propose the Sharpness-Aware Learnability (**SAL**) metric to characterize the unlearnability of parameters. We conduct experiments in both multi-task and single-task settings and find that SAL exhibits high consistency with traditional unlearnability metrics (i.e., the difference of model performance), demonstrating the rationality of SAL. Based on SAL, we further introduce Unlearnable Distance (**UD**) of samples for intuitively comparing the unlearnability of protected data. Lastly, we benchmark existing methods to reveal the current state of unlearnable examples. In summary, our contributions can be outlined as follows:

- We are the first to uncover that existing unlearnable methods fail to maintain unlearnability in multi-task models, thus offering new research directions to enhance the practicality of unlearnable examples.

- We focus on the training phase instead of simple test accuracy, and put forward an explanation for the effectiveness of unlearnable examples by analyzing the loss landscape. We further introduce Sharpness-Aware Learnability (SAL) and Unlearnable Distance (UD) as metrics for measuring the unlearnability of model parameters and data, respectively.

- Utilizing the proposed UD, we benchmark existing unlearnable methods and provide a more intrinsic tool for evaluating UEs. Our approach ascertains the gap between existing research efforts and the truly UEs while encouraging the development of more practical unlearnable methods from a novel perspective.

## 2 RELATED WORK

**Unlearnable Examples.** Data poisoning techniques that introduce perturbations to the entire training dataset, referred to as "unlearnable datasets" or simply "poisons", are also known as availability attacks (Fowl et al., 2021; Yu et al., 2022), generalization attacks (Yuan & Wu, 2021), delusive attacks (Tao et al., 2021), or unlearnable examples (Huang et al., 2021). In this study, unlearnable

datasets are considered defensive measures, as their primary purpose is to prevent the exploitation of data. Conversely, attempting to learn from unlearnable datasets is regarded as an attack.

Unlearnable datasets are generated by applying perturbations to clean samples while ensuring that the labels are unchanged. All methods for crafting UEs aim to address the following bi-level maximization problem:

$$\max_{\delta \in \Delta} \mathbb{E}_{(x,y) \sim \mathcal{D}_{\text{test}}} [\mathcal{L}(f(x), y; \theta(\delta))], \tag{1}$$

$$\theta(\delta) = \arg \min_{\theta} \mathbb{E}_{(x_i, y_i) \sim \mathcal{D}_{\text{train}}} [\mathcal{L}(f(x_i + \delta_i), y_i; \theta)]. \tag{2}$$

Equation 2 describes the process of training a model on unlearnable data, where $\theta$ denotes the model parameters. Equation 1 states that the unlearnable data should be chosen so that the trained network has high test loss, and thus fails to generalize to the test set.

**Existing Explanations.** There are multiple explanations for why unlearnable datasets hinder the generalization of networks on test sets: error-minimizing perturbations lead to overfitting (Huang et al., 2021), error-maximizing noise promotes the learning of non-robust features (Fowl et al., 2021), convolutional layers are receptive to autoregressive patterns (Sandoval-Segura et al., 2022b). These various explanations stem from different optimization objectives and theories. However, the predominant explanation is provided by (Yu et al., 2022), who discover nearly perfect linear separability of perturbations across all considered unlearnable datasets. They propose that unlearnable datasets introduce learning shortcuts as a result of the linear separability of perturbations. However, existing explanations treat the model trained on UEs as a whole and focus only one-siddely on its accuracy on a clean test set, in addition to the fact that existing explanations have difficulty explaining why UEs fail in multi-task scenarios. In addition, some studies have revealed that UEs tend to exhibit a high peak accuracy and lower final accuracy on test dataset of classification tasks (Sandoval-Segura et al., 2022a; Zhu et al., 2024). However, since learning is a dynamic process that is only completed during the training phase, it is challenging and even wrong to discern the underlying reasons for the fluctuation in accuracy numbers. To truly understand the inherence and impact of unlearnability, one must focus on the training process itself.

**Loss Landscape Sharpness.** The notion of the loss landscape sharpness and its connection to generalization has been extensively studied, both empirically (Keskar et al., 2016; Jiang et al., 2019; Neyshabur et al., 2017; Li et al., 2018) and theoretically (Dziugaite & Roy, 2017). These studies have motivated the development of methods (Chaudhari et al., 2019) that aim to improve model generalization by manipulating or penalizing sharpness. Among these methods, Sharpness-Aware Minimization (SAM) (Foret et al., 2020; Andriushchenko & Flammarion, 2022; Wen et al., 2022) has shown to be highly effective and scalable for DNNs across various tasks (Li et al., 2024). The loss landscape helps illustrate how training data is learned. Similarly, it can be employed to demonstrate how UEs are unlearned. In this paper, we start from loss landscape analysis and then borrow the use of sharpness to explain *unlearnability*.

## 3 SHARPNESS-AWARE UNLEARNABILITY EXPLANATION

### 3.1 EXPLORATION OF THE MODEL OPTIMIZATION PROCESS

**Experimental Setting.** For the toy classification task, we employ a single-layer linear layer as the classification model with input and output dimensions of $(12,)$ and $(10,)$, respectively. We manually construct a dataset containing 5,000 samples and 10 classes using the *make_classification* method in the scikit-learn library. We utilize CrossEntropy as the discriminator and SGD (with a learning rate of 0.1) as the optimizer. We train for 10 epochs and collect trajectories for plotting the loss landscape every 5 steps. For MNIST, we train a classifier using LeNet-5, with the same discriminator and optimizer (with a learning rate of 0.001 and momentum of 0.9). For both datasets, we employ OPS to generate class-wise perturbations to construct UEs.

**Visulization of Loss Landscape.** The parameters in DNNs are very high dimensional and there are a lot of nonlinearities involved, making it hard to imagine what is going on during the optimization. While much work has studied the loss landscape of deep networks in vanilla training, there has been little discussion of poisoning training, especially for training on UEs. Here we study the difference between the loss landscapes of the vanilla and poisoning training and the convergence trajectories

of the models. Note that the two parameters that need to be chosen for 2D loss landscape plotting to estimate the effect of overall model parameter changes on loss is difficult because DNNs often have millions of parameters. Therefore, the fewer model parameters, the more accurate the estimation, and the 2D loss landscape and convergence trajectories can better reflect the actual training process. Furthermore, when there are only two parameters, the loss landscape can fully and accurately represent the training process. However, this extreme case is not suitable for poison training. Therefore, we choose a single linear layer classifier with parameter dimensions of (12,10) to balance the requirements of the model's ability to learn effectively under regular training settings and the potential interference between an excessive number of parameters.

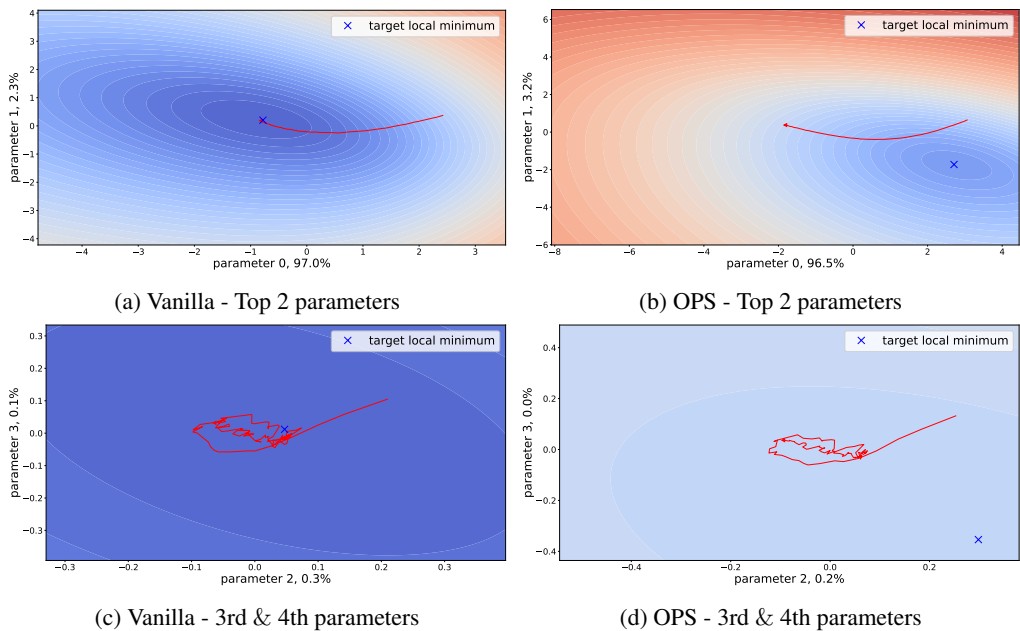

(a) Vanilla - Top 2 parameters

(b) OPS - Top 2 parameters

(c) Vanilla - 3rd & 4th parameters

(d) OPS - 3rd & 4th parameters

Figure 1: Optimization in loss landscape of training process on Toy Classification task. We use PCA for dimensionality reduction and the x, y-axis are selected **Top 2** parameters. The training campaign of the model on UEs tends to take detours rather than shortcuts to the target minimum. However, only very few parameters converge in line with the model performance. For ease of comparison, we employ the same colour bars in all figures. The fewer contour lines in the bottom two figures are due to the loss fluctuations being significantly lower than those in the top two figures. Refer to Appendix A.5 for loss landscape on MNIST dataset and more details of PCA.

To better demonstrate the degree of influence of different parameters on model convergence during model optimization, we use principal component analysis (PCA) on the optimization path and get the top 2 components (parameters) to visualize the loss over the 2 orthogonal directions with the most variance, as well as to choose the 3rd and 4th components, respectively. In Figure 1, we use a simple linear classification task with a very small number of parameters to show how UEs work. For a real-world scenario, we adopt MNIST as datasets for image classification task and the corresponding visualization is shown in Figure 10 in Appendix A.5. There are two conclusions that can be drawn from these loss landscape visualizations. Firstly, the model trained on UEs initially approaches the global optimum and then gradually deviates, indicating that the model is not yet fully affected by UEs during the early stages of poison training. This is consistent with our experimental results in Section 3.3. The second is that various parameters have varying levels of importance in model training. This leads us to speculate that only a few key parameters in DNNs undergo normal "learning" and updating during the training campaign. In simpler terms, a small subset of parameters significantly influences the model's ability to learn when trained on UEs, whereas most secondary parameters do not undergo normal "learning". Hence, we suppose that the abnormal updates of key parameters reflect this unlearnability of models, and unlearnable perturbations to the samples make it difficult to update the parameters properly. The evaluation of UEs should take into account the updates of model parameters, rather than solely relying on simple test accuracy.

## 3.2 Unveiling the Relationship between Loss Landscape and Unlearnability

The loss landscape provides an intuitive visualization of the model's optimization trajectory, illustrating how training data is learned. Similarly, it can be employed to demonstrate how UEs are unlearned. Hence, we propose to explore the relationship between Loss Landscape and unlearnability. If a sample is unlearnable, it needs to satisfy one of the following two conditions: (1) the parameter update direction moves along the contour lines; (2) the parameter update magnitude is small enough so that the training loss barely changes. Implementing the first condition is quite challenging, as parameters are usually updated in a direction perpendicular to the gradient. Therefore, we need to strive to satisfy the second condition as much as possible.

As shown in Figure 2(a), we display two regions with dense and sparse contour lines, respectively. When the parameters are in the flat area of the loss landscape, the change in loss after the parameter update is minimal, presenting a performance similar to the second condition. In contrast, in the steep area, even small parameter updates can cause the loss to decrease rapidly. Thus, we believe that the steepness of the loss landscape can reflect the unlearnability of UEs. In Figure 2(b), within a certain range of parameter updates, the fluctuations in loss in the flat area are relatively small, which inspired us to propose Sharpness-Aware Learnability. Our focus is on the relationship between the fluctuation of loss within a certain region and the perturbation of parameters. When training on UEs, the optimization direction should be as tangential to the gradient direction as possible, rather than in the opposite direction as with adversarial examples, to ensure that no useful information is learned.

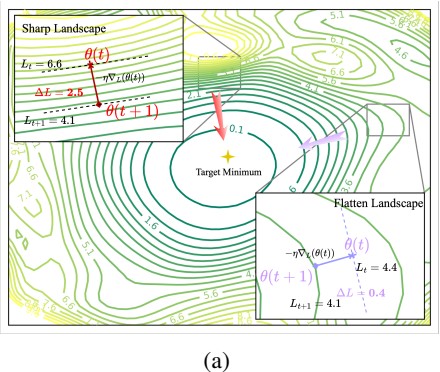
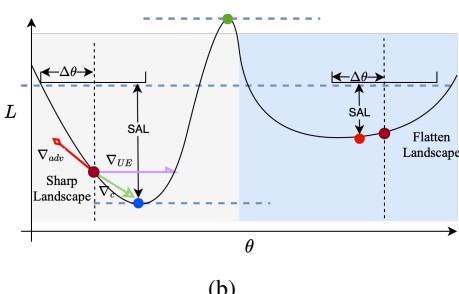

(a)            (b)

Figure 2: (a) Training loss landscape of ResNet-46. It is more difficult for the model to converge to the target optimal point in flatten regions (lower SAL) for the same step size compared to sharp regions. (b) When the neighborhood is flatter, the smaller the SAL of the model parameters is, meaning that it is difficult for the model to escape from the local minima because this requires more steps compared to when it is in a sharp neighborhood.

## 3.3 SAL for Single-task Scenarios

**Definition 1 (Sharpness-Aware Learnability, SAL).** *Inspired by (Zhu et al., 2023), in this work, we define the **S**harpness-**A**ware **L**earnability of layer parameters of model trained on particular dataset as SAL. Given a weight perturbation scaling factor $\epsilon > 0$ and a neural network $\boldsymbol{\theta}$, the SAL of layer parameters $\boldsymbol{\theta}_l$ at training epoch $t$ is defined as:*

$$SAL\left(\boldsymbol{\theta}_l, \epsilon, t\right) = \max_{\|\boldsymbol{v}\|_p \leq \epsilon} \left|\mathcal{L}(\boldsymbol{\theta}_l + \boldsymbol{v}; \mathcal{D}_{tr}) - \mathcal{L}\left(\boldsymbol{\theta}_l; \mathcal{D}_{tr}\right)\right|, \tag{3}$$

*where $\boldsymbol{\theta}$ is a $l$-layer DNN and $\boldsymbol{\theta}_l$ is the $l$-th layer parameters. $\boldsymbol{v}$ is the perturbation of $\boldsymbol{\theta}_l$, and the parameters of the remaining layers are temporarily frozen. The target training loss (e.g. cross-entropy) is denoted by $\mathcal{L}$. $\|\cdot\|_p \leq \epsilon$ is denoted as the $\ell_p$ norm. $\mathcal{D}_{tr}$ denotes the training dataset.*

Considering that all current unlearnable methods target a single task (e.g. image classification), we first focus on monitoring SAL on a single task, as well as explaining why SAL is a better explanation of why and how existing unlearnable methods work.

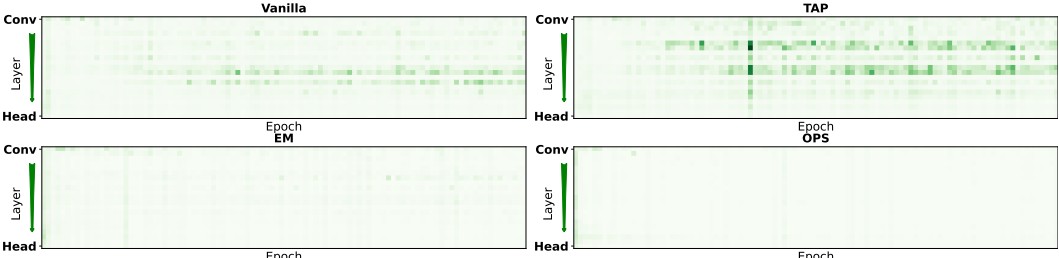

Figure 3: Most parameters in the model trained on UEs exhibit a low SAL (except for TAP, which is considered as adversarial examples), while a small number of parameters on clean training maintain a relatively high SAL (darker green and larger SAL). This suggests that UEs exert unlearnability by reducing the SAL of the parameters. Results on other UEs are in Figure 11 in Appendix A.6

**Experimental Setting.** Unless stated separately, all experiments in our graphical results are on CIFAR-10 (Krizhevsky et al., 2009) trained with ResNet18 (He et al., 2016). We generate sample-wise perturbations for all the UEs mentioned except for OPS and we follow their default generating setting. We limit the perturbation budget to $\ell_\infty = 8/255$. After generating the poisoned dataset, the model is randomly initialized and re-trained from scratch, via SGD for 100 epochs with an initial learning rate of 0.1 and decay by 0.1 by default. For SAL searching, we set $\|\cdot\|_p = \|\cdot\|_2$ and $\epsilon = 0.05$, the iterative step for optimizing $v$ is 10.

Figure 3 shows the visualization of SAL of different layer parameters as training proceeds, visualization on more UEs is shown in Figure 11. We observe that models trained on unlearnable datasets have a lower SAL compared to those trained on vanilla datasets, and the SAL of these parameters further decreases with training proceeding. Although this generally corresponds to the pattern of their demonstrated test accuracy, where methods that exhibit lower test dataset accuracy have fewer unlearnable parameters, there are also exceptions: TAP (Fowl et al., 2021) use adversarial examples for availability attack and get a significant test accuracy degradation. This helps us to distinguish which methods truly demonstrate unlearnability and which are simply conventional poisoning attacks. In summary, SAL can better explain the origins and working principles of unlearnability.

## 3.4 SAL FOR MULTI-TASK SCENARIOS

**Background and Experimental Setting.** We observe that while existing unlearnable and availability attack methods are effective across datasets, model architectures, and training methods, there is a lack of discussion regarding multi-task scenarios. Intuitively, we believe this is primarily because most of the current unlearnable data heavily relies on simple features that are easy to learn. Clearly, class-based features are only applicable to classification tasks and cross-entropy loss. Even those unlearnable datasets that do not require class features are considered to necessitate the introduction of shortcuts, but the results in Figure 6 suggest that there are conflicts between shortcuts for different tasks. In the multi-task learning experiments we train a multi-task model based on ResNet as the backbone on Taskonomy (Zamir et al., 2018) dataset (we use the tiny split to facilitate the search for EM perturbations), which includes four tasks (*Scene Cls.*, *Keyp.2d*, *Depth Euc.* and *Segm.2D* ). We refer to the experimental setup of ModSquad (Chen et al., 2023) and train for 100 epochs.

The unlearnability of EM perturbations during multi-task training is not significant. We believe that this is largely due to the inherent difficulty of multi-task learning, that is, conflicts between different tasks naturally exist, and unlearnable perturbations are also limited to this. From the perspective of loss, for samples in the same batch, whether it is adding perturbations that maximize loss (adversarial samples) or perturbations that minimize loss (EM), this is more difficult in the multi-task scenario. In addition, existing unlearnable methods often overlook the discussion of unlearnable perturbation on complex tasks such as semantic segmentation and depth estimation.

To sum up, the proposed SAL provides a more in-depth analysis of why existing methods fail in multi-task scenarios. That is, the impact of different task losses on the changes of the same parameter is heterogeneous, making it difficult for the parameter to exhibit learnability or unlearnability simultaneously across different tasks. We provide heat maps of cosine similarity between vectors

composed of SAL parameters at different layers during training for different tasks in Figure 4. It can be observed that there is no significant difference in the parameter SAL similarity between tasks in the vanilla training and EM training. Based on our theory that unlearnability is reflected through the parameters, this confirms that existing unlearnable methods struggle to demonstrate extensive unlearnability in multi-task scenarios (Yu et al., 2020).

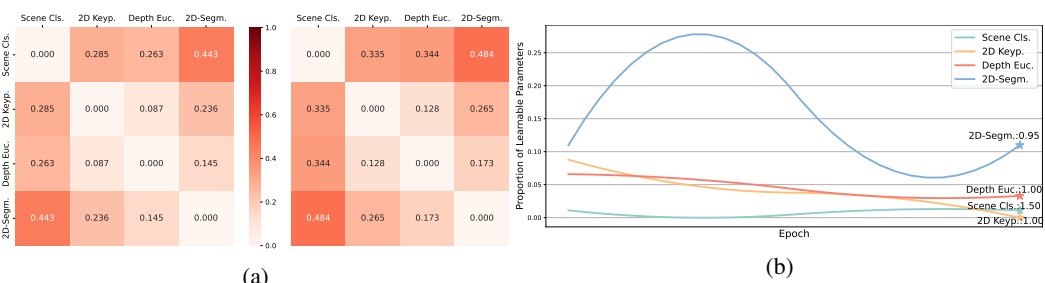

(a)                                              (b)

Figure 4: (a) Heat maps of the cosine similarity between vectors composed of SAL parameters for different tasks. A higher value implies that the unlearnability of the parameters between two tasks is more consistent during training, while a lower value suggests that the two tasks are more difficult to exhibit unlearnability simultaneously. Furthermore, we believe that the inherent conflicts present in multi-task learning are the fundamental cause of the inconsistency in unlearnability. (b) The proportions of learnable parameters for different tasks have significantly different trends in EM training, and the inconsistency of unlearnability between tasks reveal the great challenge of constructing UEs under multi-task scenarios. We label the *unlearnable distance* computed by Equation 5 at the end of the smoothed line (it is time-consuming to perform SAL analysis for multi-task models).

## 4   DATA UNLEARNABILITY METRIC: UNLEARNABLE DISTANCE

In this section, we delve further into evaluating the unlearnability of data based on the SAL. Drawing from the findings in Section 3.1 and Section 3.3, the SAL reflects the learning capacity of model parameters, meaning that when the SAL is relatively large, the parameter has a significant impact on the model's performance. Furthermore, we discovered that an unlearnable dataset leads to training failure by reducing the SAL of model parameters. In other words, the proportion of parameters with high SAL can directly indicate the unlearnability of a dataset.

Based on this insight, we propose categorizing parameters into learnable and unlearnable groups using a threshold according to the SAL. Moreover, we introduce a new metric called unlearnable distance (UD) to evaluate the unlearnability of various unlearnable methods in specific training settings. This approach allows us to assess the unlearnability of different unlearnable methods without relying solely on a single test accuracy metric. We made our code publicly available on GitHub†.

**Definition 2** (**Learnable Threshold, LT**). *The goal of LT is to distinguish between learnable and unlearnable parameters, which relies on the SAL of the model parameters during the training process of the clean model $\theta^c$. Considering the diverse density distributions of the SAL, we employ the K-Means method (Arthur et al., 2007) to segregate the parameters into two categories and choose the mean value of the cluster centers as the threshold. For the sake of convenience, we utilize $\beta$ to represent LT, and its value is determined as follows:*

$$\beta(T) = \frac{\sum_{t=1}^{T}\sum_{i}^{2}\kappa_i(SAL(\boldsymbol{\theta}^c, \epsilon, t))}{2 \times T}, \tag{4}$$

*where $T$ denotes the number of epochs; $\kappa_i$ represents the mean value of the $i$-th cluster center; $SAL(\boldsymbol{\theta}^c, \epsilon, t)$ refers to the sharpness-aware learnability of the clean model $\boldsymbol{\theta}^c$ at the $t$-th epoch.*

Next, we evaluate the unlearnability of UEs by comparing the changes in the number of learnable parameters during the training process of the same model architecture on clean samples and UEs. Figure 5 shows the process used by our threshold to categorize learnable and unlearnable parameters.

---

† `https://github.com/MLsecurityLab/HowFarAreFromTrueUnlearnability.git`

Furthermore, we quantify the proportion of learnable parameters for UEs during training, as depicted in Figure 13 in Appendix A.7. As expected, we find that the number of learnable parameters for clean training is significantly higher than for other UEs, which experience a rapid decrease in the number of learnable parameters from the beginning of training. TAP is an exception, as it is inherently adversarial examples. It can also be observed that while TAP has a similar number of learnable parameters to clean training, it is merely continuously learning erroneous features from the data.

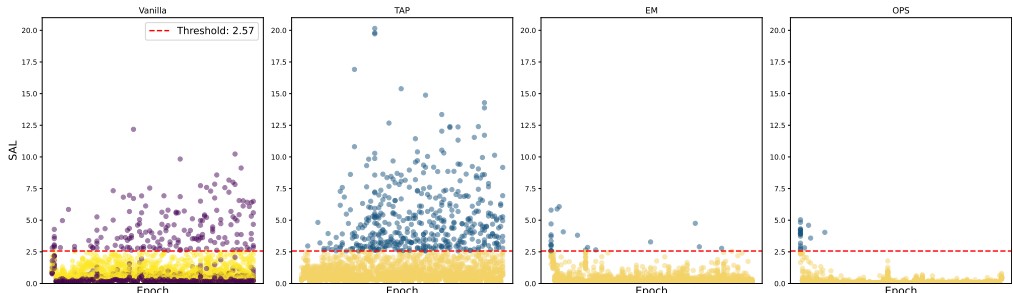

Figure 5: Finding SAL threshold $\beta(T)$ to distinguish learnable and unlearnable parameters in model trained on the vanilla dataset. The learnable parameters of poisoned models are only a small fraction in the early stages of training and decrease rapidly (except for TAP, which is considered as adversarial examples), which further corroborates our hypothesis regarding the inherent unlearnability of UEs, suggesting that the learning of part of the model parameters is compromised.

**Definition 3** (**Unlearnable Distance, UD**). *Assuming that the clean model and the poisoned model have the same network architecture, they also possess an equal number of parameters. We measure the unlearnability of the poisoned model by the ratio of the average learnable parameter quantities in the poisoned model and the clean model. If the ratio is small, it implies that the UEs contribute less to model convergence, thereby being closer to the genuine UEs. Conversely, if the ratio is large, it means that the model converges more normally, without achieving the genuine UEs. Thus, this distance can be calculated as follows.*

$$UD(\boldsymbol{\theta}^p) = \frac{\frac{1}{T^p} \sum_t^{T^p} \lambda(SAL(\boldsymbol{\theta}^p, \epsilon, t), \beta(T^c))}{\frac{1}{T^c} \sum_t^{T^c} \lambda(SAL(\boldsymbol{\theta}^c, \epsilon, t), \beta(T^c))}, \tag{5}$$

*where $T^c$ and $T^p$ represent the number of epochs for the training of the clean model and the poisoned model, respectively. $\lambda(SAL(\boldsymbol{\theta}, \epsilon, t), \beta(T^c))$ denotes the number of learnable parameters in the model $\boldsymbol{\theta}$ at the $t$-th epoch, i.e., the corresponding SAL is greater than the threshold $\beta(T^c)$.*

**Algorithm.** Here, we provide a complete summary of the process to evaluate the unlearnability of a UE dataset $\mathcal{D}^p$, as shown in Algorithm 1.

## 5 BENCHMARKING DATA UNLEARNABILITY

### 5.1 EXPERIMENTAL SETTING

The basic experimental setup in this section is almost identical to Section 3.3. To demonstrate the universality of the proposed metrics, we have included the larger-scale dataset CIFAR-100 and ImageNet (Russakovsky et al., 2015) subset (the first 100 classes, and we center-crop all the images to 224×224) in addition to CIFAR-10. Moreover, we have explored various base model architectures such as ResNet-50, SENet-18 (Cheng et al., 2016), and ViT, apart from ResNet-18.

In terms of unlearnable methods, we have selected EM (Huang et al., 2021), REM (Fu et al., 2022), DC (Feng et al., 2019), TAP (Fowl et al., 2021), LSP (Yu et al., 2022), and OPS (Wu et al., 2022) as exploration models, all of which are representative. We refer to the experimental setup in (Qin et al., 2023a) to get UEs and implement training and defense setups.

---

**Algorithm 1** Unlearnable Distance

---

**Input:** Poisoned training dataset $\mathcal{D}_{tr}^p$, Clean training dataset $\mathcal{D}_{tr}^c$, Poisoned model $\boldsymbol{\theta}^p$, Clean model $\boldsymbol{\theta}^c$, Training epochs $T^c$ on clean data, Weight perturbation scaling factor $\epsilon$, Training epochs $T^p$ on UEs.
**Output:** $UD(\boldsymbol{\theta}^p)$
1: $\boldsymbol{\theta}^p \leftarrow \boldsymbol{\theta}_0, \boldsymbol{\theta}^c \leftarrow \boldsymbol{\theta}_0$                                          ▷ Initialization
2: **for** $t$ in $1, ..., T^c$ **do**
3:      $\boldsymbol{\theta}^c(t+1) \leftarrow \boldsymbol{\theta}^c(t)$
4:      **for** the $l$-th layer parameters $\boldsymbol{\theta}_l^c$ in $\boldsymbol{\theta}^c$ **do**
5:          Freeze($\boldsymbol{\theta}_i^c$) $\forall i \neq l$
6:          $SAL^c(\boldsymbol{\theta}_l^c, \epsilon, t) \leftarrow \max_{\Delta\boldsymbol{\theta}_l \in \mathcal{C}_{\boldsymbol{\theta}}} \mathcal{R}(\boldsymbol{\theta}_l^c + \Delta\boldsymbol{\theta}_l) - \mathcal{R}(\boldsymbol{\theta}_l^c)$             ▷ Follow Equation 3
7:      **end for**
8: **end for**
9: $\beta(T^c) \leftarrow \frac{1}{2 \times T^c} \sum_{t=1}^{T^c} \sum_i^2 \kappa_i(SAL(\boldsymbol{\theta}^c, \epsilon, t))$             ▷ Calculate Learnable Threshold
10: **for** $t$ in $1, ..., T^c$ **do**
11:      $\boldsymbol{\theta}^p(t+1) \leftarrow \boldsymbol{\theta}^p(t)$
12:      **for** the $l$-th layer parameters $\boldsymbol{\theta}_l^p$ in $\boldsymbol{\theta}^p$ **do**
13:          Freeze($\boldsymbol{\theta}_i^p$) $\forall i \neq l$
14:          $SAL^p(\boldsymbol{\theta}_l^p, \epsilon, t) \leftarrow \max_{\Delta\boldsymbol{\theta}_l \in \mathcal{C}_{\boldsymbol{\theta}}} \mathcal{R}(\boldsymbol{\theta}_l^p + \Delta\boldsymbol{\theta}_l) - \mathcal{R}(\boldsymbol{\theta}_l^p)$             ▷ Follow Equation 3
15:      **end for**
16: **end for**
17: $UD(\boldsymbol{\theta}^p) \leftarrow \frac{\frac{1}{T^p} \sum_t^{T^p} \lambda(SAL(\boldsymbol{\theta}^p, \epsilon, t), \beta(T^c))}{\frac{1}{T^c} \sum_t^{T^c} \lambda(SAL(\boldsymbol{\theta}^c, \epsilon, t), \beta(T^c))}$            ▷ Follow Equation 5

---

## 5.2 BENCHMARK RESULTS

We consider three factors that influence the unlearnability of UEs, including unlearnable methods, attack methods on UEs, and model architectures. Additionally, we provide the average number of learnable parameters per layer as a reference metric, abbreviated as "#LP."

Table 1: UD of ResNet-18 trained on UEs constructed on CIFAR-10, CIFAR-100 and ImageNet subset. Bolded numbers indicate the smallest unlearnable distance, i.e., the best unlearnability.

| Unlearnable Method | CIFAR-10 | | | CIFAR-100 | | | ImageNet-100 | | |
|---|---|---|---|---|---|---|---|---|---|
| | Test Acc | #LP | UD | Test Acc | #LP | UD | Test Acc | #LP | UD |
| Vanilla | 94.11 | 3.32 | \ | 75.23 | 2.25 | \ | 69.13 | 1.86 | \ |
| EM | 26.52 | 0.62 | 0.187 | 12.34 | 0.25 | 0.111 | 1.20 | 0.02 | **0.011** |
| REM | 30.26 | 1.54 | 0.464 | 20.32 | 2.25 | 1.000 | 4.90 | 2.38 | 1.280 |
| DC | 18.51 | 1.00 | 0.301 | 54.66 | 2.24 | 0.996 | 5.00 | 3.40 | 1.828 |
| TAP | 29.85 | 5.44 | 1.639 | 33.75 | 2.73 | 1.213 | 1.20 | 4.22 | 2.269 |
| LSP | 10.23 | 1.14 | 0.343 | 2.15 | 0.84 | 0.373 | 4.40 | 3.34 | 1.796 |
| OPS | 11.98 | 0.52 | **0.157** | 10.09 | 0.02 | **0.009** | 3.30 | 2.66 | 1.430 |

**The impact of different unlearnable methods on UD is shown in Table 1**. We find that the majority of unlearnable methods yield UE samples with relatively low UD, which is consistent with test accuracy. However, the UD of TAP is significantly greater than 1, implying that the model has ample learnable parameters. This is consistent with the aforementioned visualization of the SAL and learnable parameters for TAP, implying that TAP, as an adversarial example, although an effective availability attack method, are not true UEs. This is because adversarial examples guide the model away from the correct optimization direction during training. Hence, parameters continue to update but the model does not converge. This is inconsistent with the rapid reduction and cessation of learning in the number of learnable parameters observed in models trained on UEs. Additionally, we observe that OPS has a significantly lower UD, even though its test accuracy is not the lowest. We speculate that this is because the perturbations in OPS contain strong simple features (more pronounced class-related features), providing a *shortcut* for model training, leading to a faster reduction in learnable parameters. Except for EM, all methods have relatively large UD ($> 1$). Although the ranking among different methods does not vary significantly from the previous trend (TAP still has the lowest test accuracy and the largest UD). This implies that the unlearnability of the same method behaves differently in different datasets, which further corroborates our proposed opinion that unlearnability cannot be directly linked to test accuracy. Instead, the training environment and the selected model should be taken into account. This might be because the ImageNet-100 dataset has more abundant features, making it more difficult for the model to learn useful features from perturbed samples. Nevertheless, we do not believe that *shortcut learning* can fully explain

unlearnability, as the *shortcut* is just one of the factors contributing to the unlearnability of UEs, and it happens to be the most effective method for classification tasks.

Table 2: UD of ResNet-18 trained on defended UEs. Note that ±values in red or blue indicate changes in UD relative to predefense.

| Unlearnable Method | JPEG | | Adversarial Training | | UEraser-max | | MixUp | | CutOut | |
|---|---|---|---|---|---|---|---|---|---|---|
| | #LP | UD | #LP | UD | #LP | UD | #LP | UD | #LP | UD |
| Vanilla | 2.48 | 0.747 | 7.14 | 2.151 | 4.32 | 1.301 | 3.64 | 1.097 | 3.72 | 1.120 |
| EM | 1.24 | $0.373_{+0.186}$ | 0.40 | $0.120_{-0.067}$ | 0.42 | $0.127_{-0.06}$ | 1.28 | $0.386_{+0.199}$ | 1.56 | $0.470_{+0.283}$ |
| TAP | 1.66 | $0.500_{-1.139}$ | 2.92 | $0.880_{-0.759}$ | 0.81 | $0.243_{-1.396}$ | 1.92 | $0.579_{-1.06}$ | 2.01 | $0.605_{-1.034}$ |
| OPS | 2.16 | $0.651_{+0.494}$ | 0.56 | $0.169_{+0.012}$ | 1.18 | $0.355_{+0.198}$ | 1.60 | $0.482_{+0.325}$ | 1.54 | $0.464_{+0.307}$ |

**Benchmark on defended UEs**. We calculate UD against 3 state-of-the-art defenses (JPEG compression (Liu et al., 2023), UEraser (Qin et al., 2023b) and adversarial training) and 2 commonly used data augmentation strategies (MixUP (Zhang, 2017) and CutOut (DeVries, 2017)). The result is shown in Table 2. With the exception of TAP, most defense methods can increase UD, thereby disrupting the unlearnability of UEs. Among these defense techniques, JPEG compression performs the best. As for TAP, since we consider them adversarial examples and not belonging to UEs, these additional defense perturbations disrupt the adversarial characteristics within the data. Overall, the effectiveness of the defense further demonstrates the validity of our proposed UD metric.

Table 3: UD of models with different architectures.

| Model | Vanilla | | EM | | TAP | | OPS | |
|---|---|---|---|---|---|---|---|---|
| | #LP | UD | #LP | UD | #LP | UD | #LP | UD |
| ResNet-18 | 3.32 | \ | 0.62 | 0.187 | 5.44 | 1.639 | 0.52 | **0.157** |
| ResNet-50 | 0.69 | \ | 0.25 | 0.364 | 4.17 | 1.256 | 1.13 | 0.340 |
| SENet-18 | 3.21 | \ | 0.97 | 0.302 | 5.36 | 1.614 | 0.82 | 0.248 |
| ViT | 1.15 | \ | 1.81 | 1.573 | 1.94 | 0.584 | 2.71 | 0.818 |

**Benchmark on different model architectures**. The impact of different model architectures on UD is shown in Table 3. We select ResNet series, SENet, and ViT as backbone networks. On the one hand, we find that different model architectures exhibit considerable differences in UD, ranging from a minimum of 0.157 to a maximum of 1.639. This suggests that data unlearnability is model-dependent, rather than solely dependent on the data itself. Furthermore, we find that ViT has a notably higher UD than other methods. Compared to ResNet and SENet, ViT is generally regarded as a stronger model architecture. Thus, we can conclude that the larger the number of model parameters and the stronger the model, the more difficult it is to maintain unlearnability when training on UEs. For TAP, similar to the discussion above, it exhibits an opposite trend compared to other UEs, meaning that as the model becomes more complex, the UD value actually decreases.

# 6 Conclusion

In this work, we reveal that existing unlearnable examples do not exhibit the anticipated *multi-task unlearnability*, implying that they can still contribute to enhancing the performance of multi-task models. This finding prompts us to reevaluate the true unlearnability of data. Tackling this issue from the perspective of model optimization, we propose an explanation method based on the loss landscape to shed light on the functioning of unlearnable examples. Consequently, we introduce a metric called Sharpness-Aware Learnability (SAL) to quantify the unlearnability of parameters. Furthermore, we employ SAL to distinguish between learnable and unlearnable model parameters and propose the Unlearnable Distance (UD) as a means to quantify data unlearnability. By creating a benchmark using various unlearnable methods based on the UD metric, we aim to foster community awareness regarding the effectiveness of existing unlearnable methods.

Acknowledgments

This work is partially supported by the NSFC for Young Scientists of China (No.62202400) and the RGC for Early Career Scheme (No.27210024). Any opinions, findings, or conclusions expressed in this material are those of the authors and do not necessarily reflect the views of NSFC and RGC.

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

# A  APPENDIX

## A.1  VISUALIZATION OF UEs UNDER MULTI-TASK SCENARIOS

Please refer to Figure 7 for more details.

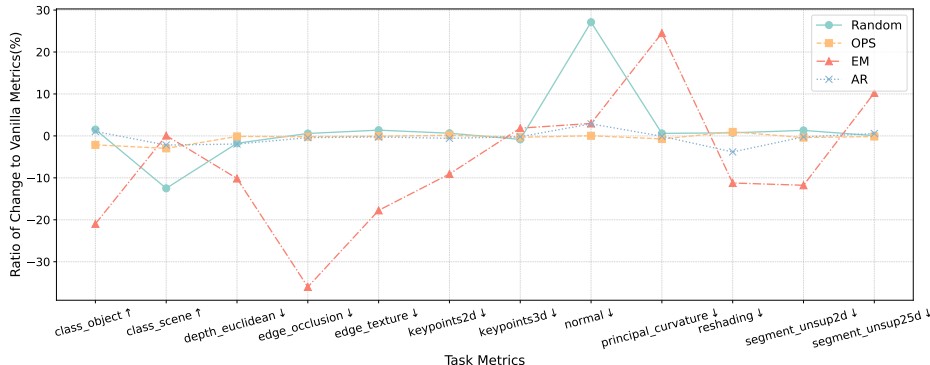

Figure 6: Current UEs fail in multi-task model training with ResNet backbone and Taskonomy dataset (the task metrics do not show a significant decrease compared to the vanilla training, and the fluctuation range is even close to random noise). The perturbations of the 4 selected UEs are considered to have obvious linear separability in classification; yet in multi-task practice, they do not even succeed in the classification tasks. The results demonstrate that existing UEs struggle to perform effectively in multi-task scenarios.

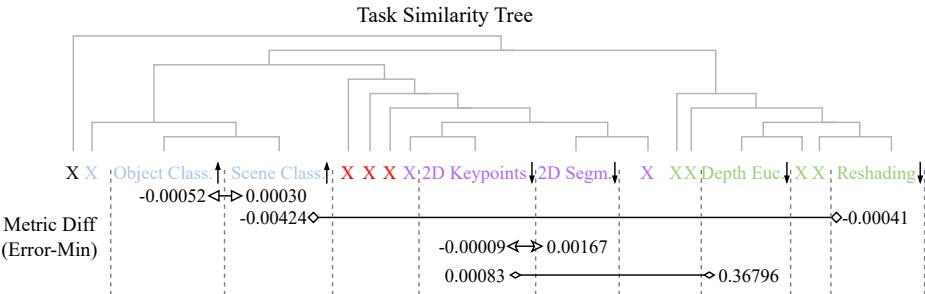

Figure 7: The visualization of the performance of EM (Error-Minizing) noise under multi-task learning. The task similarity tree in the figure is derived from Zamir et al. (2018), where **X** represents other tasks. The task metric values and connections in the figure represent the impact of perturbations of UEs found through EM training (with the loss function being the sum of the tasks at both ends of the connection) on vanilla training across tasks of different similarity. It can be observed from the figure that perturbations found using the EM are difficult to succeed, whether for tasks with high similarity (*Object Class.* and *Scene Class.*) or for tasks with low similarity (*Scene Class.* and *Reshading*). The experimental results imply that the failure of UEs on multi-task learning is not only due to conflicts between different tasks but may also be because the perturbations are more fragile than the samples, with fewer distinct features.

## A.2  VISUALIZATION OF UEs UNDER CROSS-TASK SCENARIOS

Please refer to Figure 8 for more details.

## A.3  VISUALIZATION OF THE CUMULATIVE DISTRIBUTION OF MODEL PARAMETERS

Please refer to Figure 9 for more details.

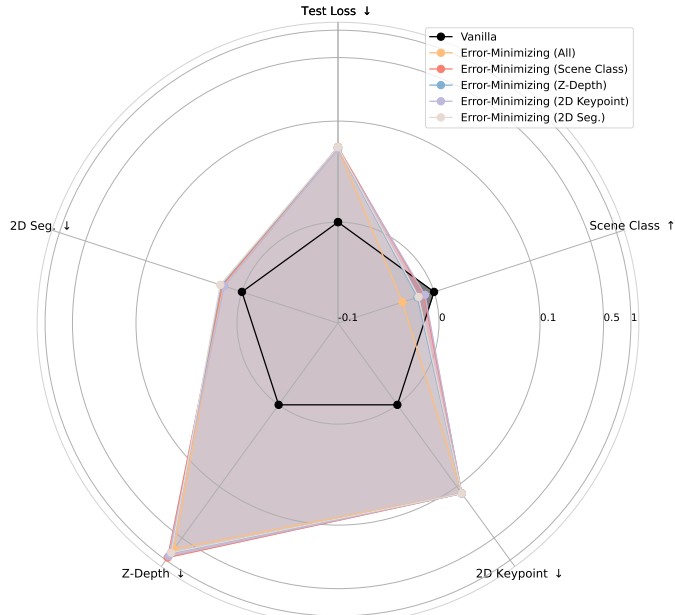

Figure 8: Difference in model performance between poisoned models (trained on UEs) and clean models (trained on the vanilla dataset) under cross-task scenarios with ResNet backbone and Taskonomy dataset. EM (∗) denotes the Error-Minimizing (Huang et al. (2021)) perturbation using the loss function of a specific task, whereas **All** signifies the use of the sum of loss functions from all tasks. The ↓ following the metric indicates that a lower value corresponds to better model performance. The results demonstrate that existing UEs struggle to perform effectively in cross-task scenarios.

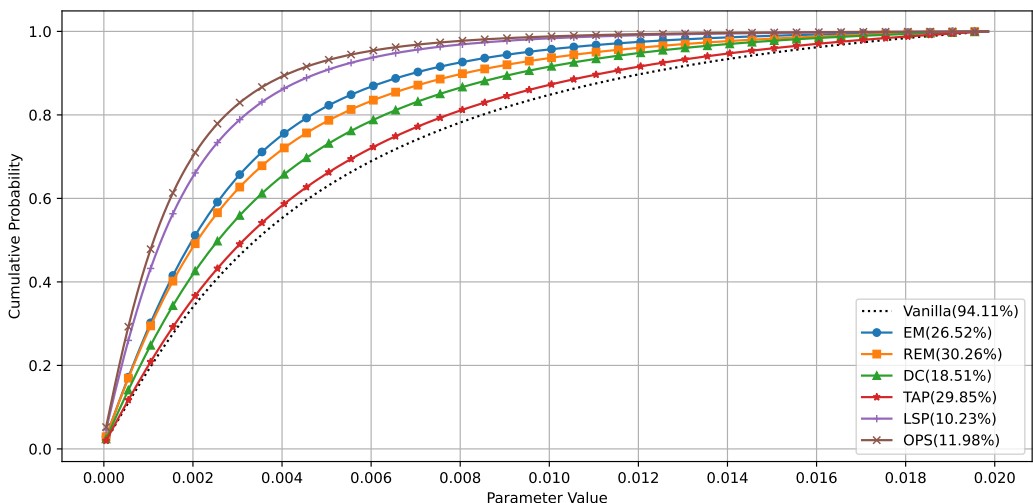

Figure 9: Visualization of the CDF of model parameters between clean training and poison training. It can be clearly seen that the distribution of the parameter values of the model trained on UEs is smaller than vanilla training.

## A.4 DETAILS OF PCA IN VISUALIZING OPTIMIZATION TRAJECTORY

Visualization of loss landscape can help us feel the way of the learning campaign of models ( Li et al. (2018)). In the high dimensional parameter space, the most straightforward way is to find two directions to cut through the high dimensional space and visualize loss values over that plane. The path's

projection onto the plane spanned by 2 random vectors in high dimensional space will look like a random walk because they have a high probability of being orthogonal, and can hardly capture any variation for the optimization path. Therefore, there are two approaches to better visualize the trajectory of model optimization. The first approach involves employing PCA along the optimization path to identify the two most relevant orthogonal directions. The second approach aims to reduce the number of model parameters as much as possible to compress the dimensionality space. We utilize the PCA method from the sklearn library, setting $n\_components = min(optim\_path\_matrix.shape)$. Our findings reveal that whether it is the LeNet-5 trained on the MNIST dataset or a simple classifier with parameter dimensions of (12,10) trained on a Toy Classification task, the first two parameters account for over 90% in the PCA analysis. This implies that the relevance of subsequent parameters is significantly lower, making it challenging to effectively visualize the optimization trajectory.

### A.5 VISUALIZATION OF LOSS LANDSCAPE

This is the visualization of the loss landscape of the training process on the naive classification dataset and MNIST dataset.

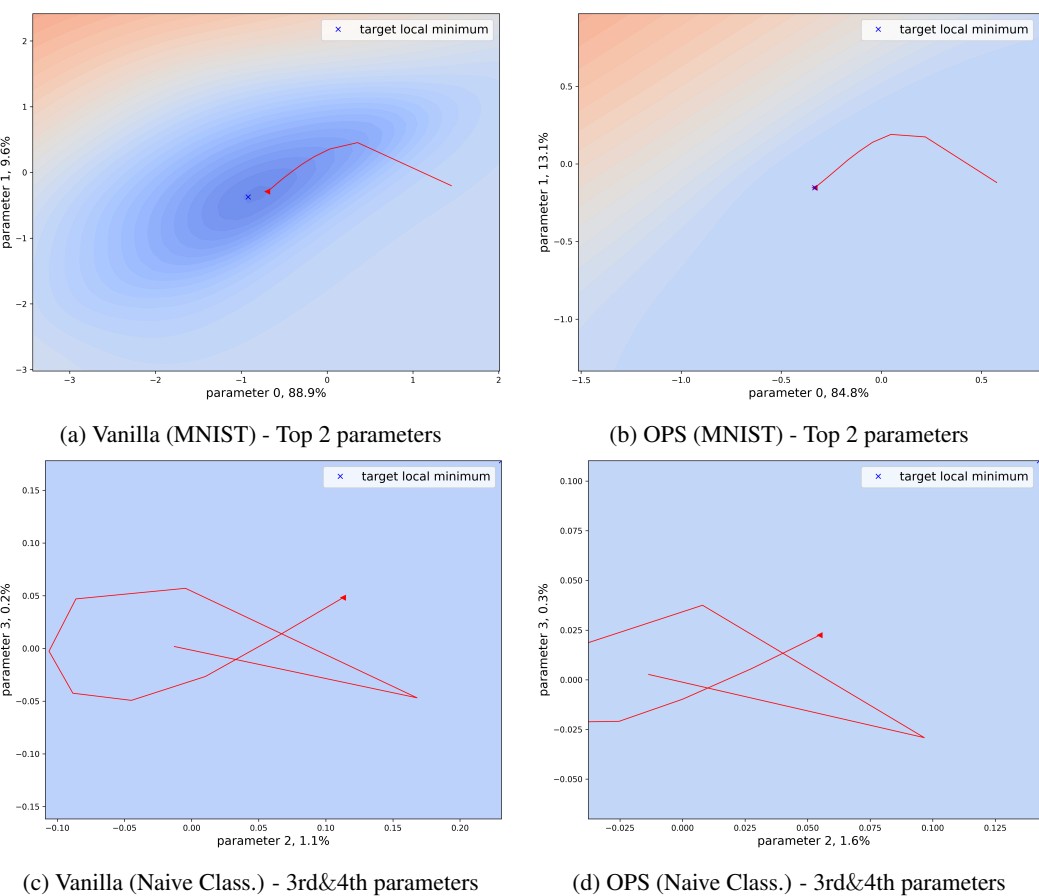

(a) Vanilla (MNIST) - Top 2 parameters

(b) OPS (MNIST) - Top 2 parameters

(c) Vanilla (Naive Class.) - 3rd&4th parameters

(d) OPS (Naive Class.) - 3rd&4th parameters

Figure 10: Optimization in loss landscape of training process on naive classification dataset and MNIST dataset. We use PCA for dimensionality reduction and the x-axis and y-axis are selected 3rd&4th parameter values.

### A.6 VISUALIZATION OF SAL

This is an additional visualization of the results for Figure 3.

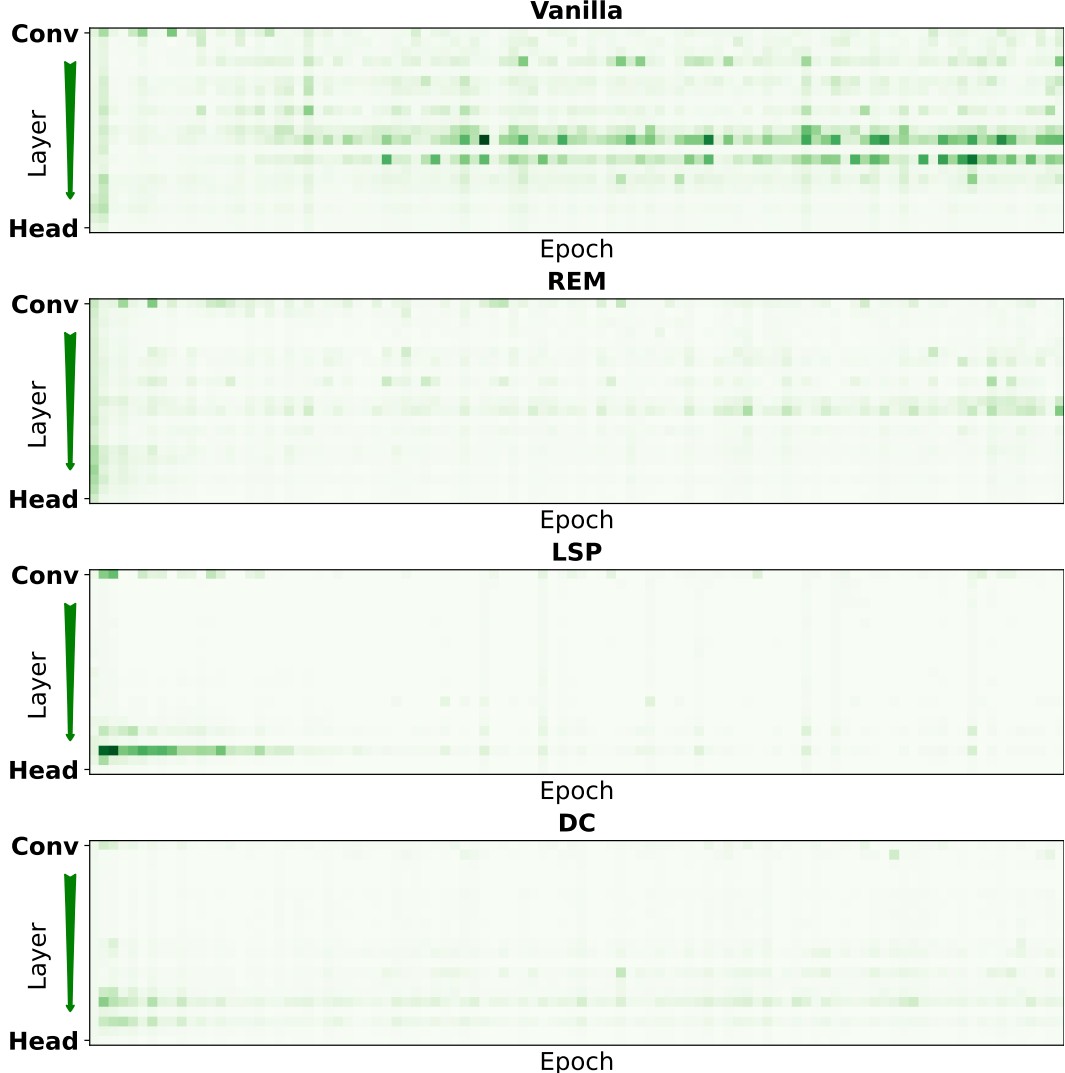

Figure 11: Heat map of SAL variation of model parameters with training epochs.

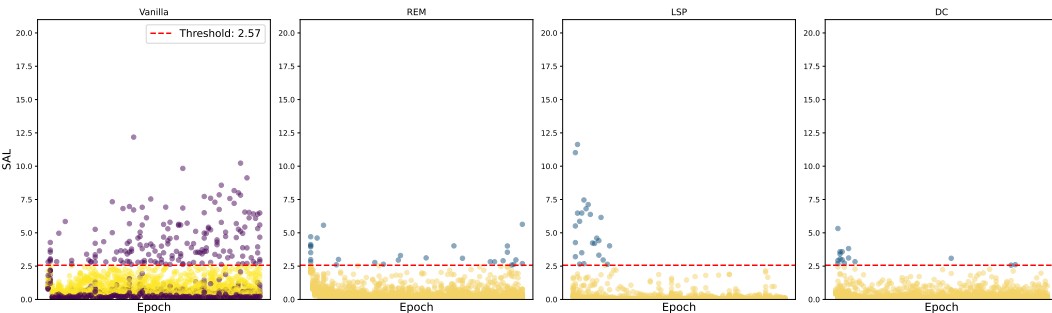

Figure 12: Finding SAL threshold $\beta$ to distinguish learnable and unlearnable parameters on the vanilla dataset by K-means. The poisoned models have only a small number of learnable parameters in the early training stage, which quickly diminish as the training proceeds.

## A.7  VISUALIZATION OF LEARNABLE PARAMETERS

This is a visualization of the proportion of learnable parameters under different UEs with SAL threshold $\beta$ obtained from Equation 4.

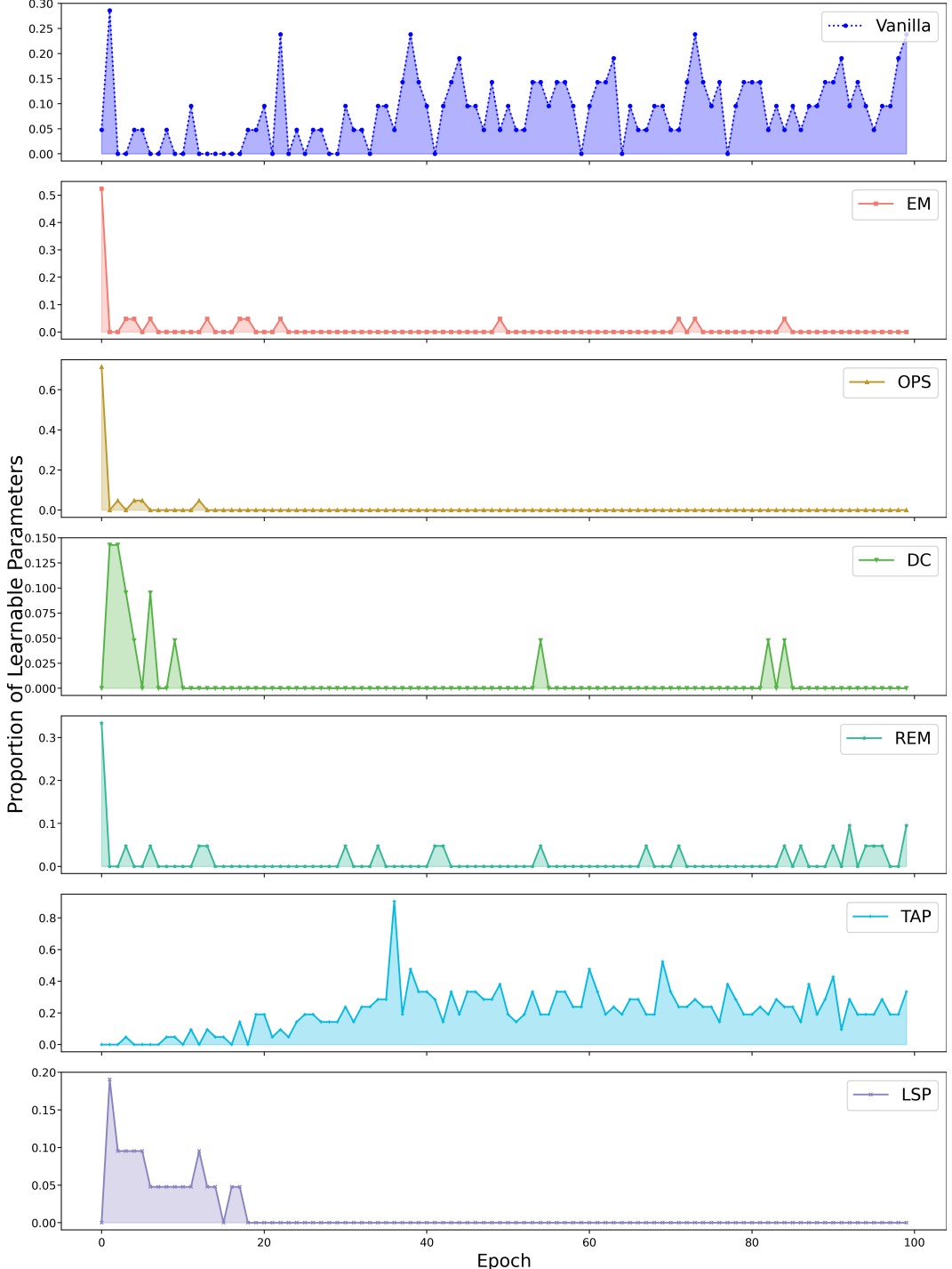

Figure 13: Proportion of learnable parameters under different UEs with SAL threshold $\beta$ obtained from Equation 4.

