# OpenReview forum: "How Far Are We from True Unlearnability?"
_ICLR.cc/2025/Conference — ICLR 2025 Poster_

### Official Review · Reviewer_yDjs · 2024-10-29

**Soundness:** 3
**Presentation:** 3
**Contribution:** 3
**Rating:** 6
**Confidence:** 3

**Summary:**

This paper addresses the concept of “unlearnable examples” (UEs), which are data samples designed to prevent unauthorized model training by impairing their usability. While current methods produce UEs effective in single-task settings, the paper reveals their limitations in multi-task scenarios. The authors explore unlearnability through the lens of model optimization and introduce Sharpness-Aware Learnability (SAL) as a metric to measure parameter unlearnability. Additionally, they propose “Unlearnable Distance” (UD) to benchmark existing UE methods across tasks.

**Strengths:**

1. The paper identifies a crucial gap in current unlearnable example (UE) methods by showing their limited effectiveness in multi-task scenarios.
2. The authors present SAL as a metric that evaluates the unlearnability of parameters, providing a fresh approach that goes beyond traditional test accuracy to measure how model parameters respond to UEs during training.

**Weaknesses:**

While the paper highlights the limitations of existing unlearnable example (UE) methods in multi-task scenarios, it does not provide solutions or adaptations to improve UE robustness across tasks.

**Questions:**

Given the identified limitations of existing unlearnable examples (UEs) in multi-task scenarios, could SAL and UD metrics be used to improve UE robustness across diverse tasks?

---

> ### Author Response · Authors · 2024-11-23
>
> Thanks for the insightful comments and evaluation.
> ___
> (1) *While the paper highlights the limitations of existing unlearnable example (UE) methods in multi-task scenarios, it does not provide solutions or adaptations to improve UE robustness across tasks.*
>
> We indeed do not propose a method for constructing multi-task UEs in this paper for several reasons:
> 1. The primary contribution of our paper  lies in identifying issues with existing UE methods from a multi-task perspective, analyzing the essence of unlearnability, and introducing a novel quantitative metric combined with model parameters to assess the non-learnability of datasets;
> 2. Due to the unknown training objectives, multi-task UE is clearly a greater challenge in the practical realm of data privacy protection, which has not yet been emphasized by current scholars, thus we leave this challenge for future discussions and endeavours;
> 3. Nonetheless, we have offered two new insights to inform future research on multi-task UEs:
>  (1) construction of UEs should not be decoupled from the model itself, as the manifestation of sample unlearnability stems from abnormal updates during the model parameter optimization process, with samples predating model selection in the training process. Therefore, the construction of UEs should consider the underlying training objectives and the model;
>  (2) the construction of multi-task UEs should focus more on the feature extraction from the model backbone; only if the feature extraction of multi-task learning is affected can the performance of the model's multi-task heads be further impacted, thereby demonstrating unlearnability.
> 4. For other practical impacts of our research, please see our No.(2) response to Reviewer 6Uf4.
> ___
> (Q1) *could SAL and UD metrics be used to improve UE robustness across diverse tasks?*
>
> Yes, as mentioned in the above response, the key to multi-task UEs lies in their impact on the extraction of shared features among multiple tasks. Most of the parameters in a multi-task model are located in the backbone, and the influence of UEs on neurons at different layers and positions varies. By employing the SAL and UD metrics we propose, we can analyze the unlearnability of UEs from the perspective of parameter updates, rather than solely relying on simple task metrics, which is clearly insufficient.

---

> ### Author Response · Authors · 2024-11-23
> **Please consider increase your Rating and Confidence**
>
> If our answers and explanations about the contribution of our study to future multi/cross-task UEs research addressed your concerns, we sincerely hope that you will consider raising your **Rating** and **Confidence**. If you still have any doubts, we would be more than happy to discuss them with you.

---

> > ### Comment · Reviewer_yDjs · 2024-11-26
> >
> > I have read the author's response, and I maintain my rating and confidence.

---

> > > ### Author Response · Authors · 2024-11-26
> > >
> > > Thank you again for your dedication and feedback. We will continue to polish this paper.

---

### Official Review · Reviewer_H1LC · 2024-11-01

**Soundness:** 3
**Presentation:** 3
**Contribution:** 3
**Rating:** 6
**Confidence:** 2

**Summary:**

This paper addresses the challenges of unlearnable examples (UEs) in the context of model training, particularly highlighting their unexpected performance in multi-task Taskonomy. The authors reveal that existing unlearnable methods fail to achieve true unlearnability, prompting an exploration of the convergence differences between clean and poisoned models through loss landscape analysis. The authors introduce two key concepts: Sharpness-Aware Learnability (SAL) for quantifying parameter unlearnability, and Unlearnable Distance (UD) for measuring data unlearnability. The paper benchmarks existing methods using UD, aiming to illuminate the limitations of current approaches and guide future research towards more effective unlearnable examples.

**Strengths:**

- The paper is  well written and easy to follow.
- The motivation and background is explained clearly. Arguments and claims are supported by  experimental results.
- The paper provides a new perspective  to  evaluate the data unlearnability through the introduction of SAL and UD. These metrics provide valuable insights into the effectiveness of unlearnable methods and address a gap in existing evaluation methods that rely solely on post-training performance metrics

**Weaknesses:**

Limited Scope of Experiments: While the paper investigates unlearnability in multi-task settings, the experiments primarily focus on specific datasets, such as CIFAR-10 and CIFAR-100. It would be valuable to see results on larger and more complex datasets, like ImageNet.

**Questions:**

- Revisiting Figure 1, it highlights that the Error Minimization (EM) method struggles to perform effectively in multi-task scenarios. I'm curious if any recent methods have also faced similar limitations.
- I'm curious about the  generalizability of the evaluation metrics.  The experiments predominantly emphasize classification tasks; including evaluations on other downstream tasks, such as semantic segmentation, object detection, and even applications in large language models, would enhance the study's relevance and generalizability. This narrow focus may limit the applicability of the findings to other domains or unlearnable methods not considered in this research.

---

> ### Author Response · Authors · 2024-11-23
>
> Thank you for the insightful comments and evaluation, and sorry for the late response because we have been fully engaged in supplementing new experiments.
> ___
> (1) *It would be valuable to see results on larger and more complex datasets, like ImageNet.*
>
> Thanks for the suggestions. We have added results on ImageNet-100. The majority of unlearnable methods perform similarly on the ImageNet-100 dataset as they do on the CIFAR dataset. For instance, EM significantly outperforms REM, despite its perturbations lacking sufficient robustness. Notice that OPS exhibits the largest UD on ImageNet-100 accompanied by an increasing \#LP as epochs grow, showing a weirdly opposite trend to the CIFAR dataset. We leave this observation for future discussions.
>
> | Unlearnable Method | Test Acc | #LP   | UD    |   |
> |--------------------|----------|-------|-------|---|
> | Vanilla            | 69.13    | 3.10  | \     |   |
> | EM                 | 1.25     | 0.42  | 0.136 |   |
> | REM                | 4.90     | 2.50  | 0.805 |   |
> | LSP                | 5.33     | 1.83  | 0.590 |   |
> | OPS                | 5.73     | 4.37  | 1.406 |   |
>
> The full results have been added to Table 1 in the revised manuscript (experiments of a small number of UE methods have not yet been completed, and we will update them as soon as they are completed; these do not affect the analysis of the overall experimental results).
> ___
> (Q1) *Revisiting Figure 1, it highlights that the Error Minimization (EM) method struggles to perform effectively in multi-task scenarios. I'm curious if any recent methods have also faced similar limitations.*
>
> We apologize for the lack of clarity in Figure 1 of our previous submission. We have redrawn Figure 1 based on new experimental results: we include two additional popular UE methods besides EM (OPS and AR) and also introduce random noise for comparison (the perturbations for OPS and AR are based on classification tasks). We selected 12 different tasks for our analysis. The figure now demonstrates that existing UEs do not perform well in multi-task scenarios. This is not only because previous UE methods are primarily designed for simple classification tasks, but also due to the conflicts between multiple tasks and the fact that multi-task UEs noise features (with perturbation less than 8/255) are more challenging for models to learn compared to sample features.
> Furthermore, it should be noted that regarding cross-task scenarios—where UEs crafted for one task are tested for their transferability to other tasks—please refer to Appendix A.2, which illustrates that existing UEs also fail in cross-task contexts. We apologize for any confusion caused by our previous unclear distinction between cross-task and multi-task scenarios.
>
> (Q2) *generalizability of the evaluation metrics.*
>
> Although our primary experiments on the proposed metrics are conducted on classification tasks, we also conduct experiments on multi-task scenarios (see Section 3.4). The results indicate that the SAL across different tasks does not maintain a consistent trend, providing a deeper explanation for the difficulty in making multi-task UEs work effectively. Moreover, our proposed metrics are not specific to classification tasks; we calculate them using Sharpness analysis, which only requires the computation of loss and is independent of classification tasks.
> Nevertheless, your concern does make sense. There is a scarcity of research on existing UE methods for tasks beyond classification, which makes it challenging to conduct more comprehensive experimental analyses for other domains. Therefore, we also look forward to a more comprehensive dataset and benchmark that includes multi-task UEs to evaluate their unlearnability across various tasks.

---

> ### Author Response · Authors · 2024-11-23
> **Please consider increase your Rating and Confidence**
>
> We hope that our additional experiments on the ImageNet-100 and Taskonomy datasets, along with our explanations regarding the generalizability of the proposed metrics (SAL&UD), will enhance the completeness of our research. If your concerns have been addressed, please consider raising your rating&confidence. Should you have any further questions, we are more than willing to respond and engage in discussion.

---

> > ### Comment · Reviewer_H1LC · 2024-11-25
> >
> > I have read the author's response, and I maintain my rating and confidence.

---

> > > ### Author Response · Authors · 2024-11-25
> > >
> > > Thank you again for your dedication and feedback. We will continue to polish this paper.

---

### Official Review · Reviewer_6Uf4 · 2024-11-02

**Soundness:** 2
**Presentation:** 2
**Contribution:** 3
**Rating:** 6
**Confidence:** 3

**Summary:**

The authors investigate the limitations of current unlearnable examples (UEs), which aim to protect data ownership by making data unusable for model training. They discover that UEs generated for one task (e.g., image classification) can still contribute to performance improvement in other tasks (e.g., semantic segmentation) within multi-task learning settings. To address this, they introduce the concept of Sharpness-Aware Learnability (SAL) as a new metric to measure the unlearnability of model parameters. They further propose Unlearnable Distance (UD), a metric built on SAL, to quantify the unlearnability of data. Using UD, they benchmark existing unlearnable methods, revealing the gap between current research and the goal of creating truly unlearnable examples.

**Strengths:**

1. Novel perspective on unlearnability: The authors explore the concept of cross-task unlearnability. They highlight the limitations of existing unlearnable methods that primarily focus on single-task scenarios, revealing their ineffectiveness in multi-task settings.

2. Focus on the training process: Instead of relying solely on test accuracy, the authors analyze the model optimization process during training to understand how UEs impact learning. This may provide a more in-depth understanding of the unlearnability mechanism compared to previous studies that focused on post-training model performance.

3. Introduction of new metrics: The authors propose two novel metrics:

    a. Sharpness-Aware Learnability (SAL): This metric quantifies the unlearnability of parameters by considering the impact of parameter updates on the loss landscape during training.

    b. Unlearnable Distance (UD): This metric, built on SAL, measures the overall unlearnability of data by analyzing the proportion of learnable parameters in models trained on UEs compared to models trained on clean data.

**Weaknesses:**

1. Limited explanation of multi-task unlearnability failure: While the authors identify the failure of existing UEs in multi-task scenarios, they provide a limited explanation for this phenomenon. They point towards the inherent difficulties of multi-task learning, specifically conflicts between different tasks, as a potential reason. However, a more detailed analysis of how these conflicts affect unlearnability is lacking. For instance, do UEs designed for one task interfere with the learning of other tasks? Or do the shared representations learned in multi-task models somehow overcome the unlearnability effect of UEs? The authors do not fully address these questions.

2. Limited scope of multi-task experiments: The multi-task experiments are primarily conducted on the Taskonomy dataset using only 4 tasks, which might not be representative of all multi-task learning scenarios. Exploring the effectiveness of UEs on other multi-task datasets with more diverse tasks and complexities would strengthen the claim of cross-task unlearnability failure.

3. Lack of discussion on practical implications: The authors briefly mention the need for better data protection strategies, but they do not discuss the practical implications of their findings in detail. How can the insights from this research be translated into real-world solutions for protecting data ownership? What are the potential challenges in implementing these solutions?

4. Poor visualization of figures: Some figures are not well-visualized, especially the color, linewidth and font size are not well-chosen. For example, figures 1, 2, and 4 are hard to read in the printed version.

**Questions:**

Please refer to the weakness.

---

> ### Author Response · Authors · 2024-11-23
>
> Thanks for the insightful comments and evaluation, and sorry for the late response because we have been fully engaged in supplementing new experiments.
> ___
> (1) *Limited explanation of multi-task unlearnability failure and Limited scope of multi-task experiments.*
>
> Thanks for your suggestion. We have included additional experimental results and explanations regarding the failure of multi-task UEs (see Figure 1 in the Introduction and Figure 7-8 in the Appendix). We believe this is crucial because the primary objective of this paper is to explore the nature of UEs. The failure of existing UEs in multi-task scenarios has provided us with valuable insights, guiding us to understand the essence of UEs from the perspective of parameter learning. Given the unknown training objectives, data privacy protection based on UEs must overcome the challenges posed by multi-task scenarios to be effective in practice. We have provided two insights for future research aimed at addressing multi-task UEs, as detailed below:
> 1. construction of UEs should not be decoupled from the model itself, as the manifestation of sample unlearnability stems from abnormal updates during the model parameter optimization process, with samples predating model selection in the training process. Therefore, the construction of UEs should consider the underlying training objectives and the model;
> 2. the construction of multi-task UEs should focus more on the feature extraction from the model backbone; only if the feature extraction of multi-task learning is affected can the performance of the model's multi-task heads be further impacted, thereby demonstrating unlearnability.
> ___
> (2) *Lack of discussion on practical implications.*
> Advancing the practicality of unlearnable data is our common goal, and it is also the starting point of our work. Let's explore the practical impact of our research from two angles.
> - **For Current Work: Unmasking the Critical Gap in UE Methods.**
> Our study highlights the limitations of existing UE methods and underscores the importance of cross/multi-task unlearnability for data protection. Essentially, current UEs are ineffective when deployed, and we seek to alert data protectors to these inherent limitations. Additionally, we aim to provoke significant attention, calling for a comprehensive reevaluation of how UEs handle cross/multi-task unlearnability.
> - **For Future Work: Providing an Effective Tool for Evaluating UE Methods.**
> We provide a powerful tool to assess the practical unlearnability of UE methods, introducing two metrics (i.e., SAL and UD) during training. These metrics help data protectors and researchers predict UE effectiveness, minimizing resource waste. Moreover, SAL and UD could guide the development of more effective UEs by highlighting the essential characteristics they should possess.
> ___
> (3) *Poor visualization of figures.*
>
> Thanks for pointing out the issue with the visualization of figures. To address this issue, we have revised our figures to enhance their readability and clarity. For instance,
> - In Figure 1, we have **redrawn** Figure 1 based on new experimental results with three popular UE methods (EM, OPS and AR) and random noise for comparison (perturbations for OPS and AR are based on classification tasks). We select 12 different tasks. We can see that existing UEs fail in multi-task scenarios.
> - In Figure 2, we adjust the size of the displayed loss landscape region to more clearly reflect the process of model convergence.
> - In Figure 4, we increase the font size. This image reflects the sharpness of the parameters in the model trained by different UEs. The clean training and TAP are clearly darker, while the number of dark grids in other UEs decreases rapidly after a few epochs.
>
> Please see Figures 1, 2 and 4 in our revision for details. We believe these changes will enhance the clarity and effectiveness of our presentation.

---

> ### Author Response · Authors · 2024-11-23
>
> ___
>
> (Q1) *For instance, do UEs designed for one task interfere with the learning of other tasks? Or do the shared representations learned in multi-task models somehow overcome the unlearnability effect of UEs?*
>
> Thanks for the suggestion, our previous manuscript does lack detailed explanations regarding the failure of multi-task UEs. We have now supplemented the main text and appendix with additional details and explanations on this matter:
> 1. Regarding the transferability of single-task UEs, we summary them as cross-task UEs (sorry for the confusion between cross-task UE and multi-task UE). We conduct ablation experiments based on EM to verify this (see Appendix A.2). Results show UEs from single tasks struggle to transfer to other tasks, and are even ineffective within their own tasks(in multi-task models);
> 2. For multi-task UEs, we add Figure 1 and Appendix A.1 to illustrate that existing UEs fail in multi-task scenarios. This is because the parameter trends for different tasks are difficult to align consistently, even though the multi-task heads share features extracted from the same backbone. The divergence in loss functions leads to inconsistent trends, and this more fundamental issue poses a significant challenge for multi-task UEs.
> ___
> (Q2) *How can the insights from this research be translated into real-world solutions for protecting data ownership? What are the potential challenges in implementing these solutions?*
>
> Please see the above No.(2) response.

---

> ### Author Response · Authors · 2024-11-23
> **Please consider increase your Rating and Confidence**
>
> If our experiments and explanations regarding the failure of multi-task UEs, as well as our discussion on the practical implications of our contributions, have alleviated your concerns. If our polishing of the paper's presentation meets your expectations, we kindly request you to consider raising your rating. Should you have any other doubts or dissatisfaction with the presentation of the paper, we are eager to discuss and rectify them at the earliest opportunity.

---

> ### Author Response · Authors · 2024-11-26
>
> We have revised the paper again according to the comments, and the edits have been highlighted in **BLUE**:
>  1. We update the contributions in the Intro to  give a clearer summary of the practical implications that the reviewer mentioned:
>      - Utilizing the proposed UD, we benchmark existing unlearnable methods and provide a more intrinsic tool for evaluating UEs. Our approach ascertains the gap between existing research efforts and the truly UEs while encouraging the development of more practical unlearnable methods from a novel perspective.
>  2. We add more explanation in Section 3 (SHARPNESS-AWARE UNLEARNABILITY EXPLANATION) about loss landscape of UEs:
>      - Hence, we suppose that the abnormal updates of key parameters reflect this unlearnability of models, and unlearnable perturbations to the samples make it difficult to update the parameters properly. The evaluation of UEs should take into account the updates of model parameters, rather than solely relying on simple test accuracy.
>  3.  We correct many typos and expression issues again for a better presentation.
>
> Should you have any further questions or suggestions, we will be more than happy to discuss with you and implement improvements accordingly.

---

> > ### Comment · Reviewer_6Uf4 · 2024-11-26
> >
> > I appreciate the authors' response and will raise my score after reviewing the rebuttal.

---

> > > ### Author Response · Authors · 2024-11-26
> > >
> > > Thank you very much for your feedback and for raising the score!  We will keep polishing our manuscript. If you have any other suggestions and comments, feel free to discuss them with us!

---

### Author Response · Authors · 2024-11-23

We thank all reviewers for the constructive and insightful efforts in evaluating this work. We have uploaded revised files, with several modifications:
1. two new image about performance of existing UEs in multi-task scenarios (one in Introduction, one in Appendix);
2. more experimental results in larger dataset (Imagenet-100) and other multi-task UEs (on taskonomy dataset) with some new findings;
3. a clearer and more complete description of the contributions to our findings and proposed indicator UD.

The above points are marked in blue (in both main paper and supplementary materials).

Besides these points, we have also revised typos, references, figure captions, and all other minor points mentioned in the reviewing process.

Thanks again for the constructive efforts in the comments and reviews.

Author

---

### Author Response · Authors · 2024-12-01
**Rebuttal and Revision Summary**

**We sincerely thank you for your insightful feedback and constructive suggestions, which have significantly improved our manuscript. Revisions are highlighted in blue in the manuscript. Below, we summarize the key revisions and rebuttal made to address the reviewers' comments and ensure alignment with the conference's standards**:
___
### **1. Additional Experiments on Multi/Cross-Task UEs**
- **Multi-Task UEs**: We add new experiments with three popular UE methods (EM, OPS and AR) and random noise for comparison on 12 different tasks, as shown in Fig. 1 which shows that existing UEs fail in multi-task scenarios.
- **Cross-Task UEs**: Ablation experiments on 4 common CV tasks with EM perturbations (see Appendix A.2). Results show UEs from single tasks struggle to transfer to other tasks and are even ineffective within their own tasks(in multi-task models).

### **2. Additional Experiments on ImageNet-100 Dataset**
We have added results on ImageNet-100 of our proposed UD and SAL metrics. The majority of unlearnable methods perform similarly on the ImageNet-100 dataset as they do on the CIFAR dataset.

### **3. More Explanation and Analysis of Failure of Existing UEs**
- It is not surprising that existing UEs fail in multi/cross-task scenarios, not only because many of them (AR, OPS, LSP) are based on **class-related linearly separable noise** (which obviously fails in non-classification tasks), but also because the multi-task learning training process (dedicated to **resolving conflicts between tasks**) can further **undermine the unlearnability** of existing UEs.
- Essentially, based on our research findings, practical UEs need to influence the update of model parameters (specifically, **the proportion of learnable parameters**), and the changes in model parameters caused by UEs **have inconsistent trends in their impact on the loss of different tasks**.
- Given the unknown training objectives, data privacy protection based on UEs must overcome the challenges posed by multi-task scenarios to be effective in practice. We have provided two insights for future research aimed at addressing multi-task UEs, as detailed below:
   - The construction of UEs should not be decoupled from the model itself, as the manifestation of sample unlearnability stems from abnormal updates during the model parameter optimization process, with samples predating model selection in the training process. Therefore, the construction of UEs should consider the underlying training objectives and the model;
   - The construction of multi-task UEs should focus more on the feature extraction from the model backbone; only if the feature extraction of multi-task learning is affected can the performance of the model's multi-task heads be further impacted, thereby demonstrating unlearnability.

### **4. More Discussion on Practical Implications of Our Study**
We provide and summarize the practical implications of our study from two aspects:
- **For Current Work**: Unmasking the Critical Gap in UE Methods. Our study highlights the limitations of existing UE methods and underscores the importance of cross/multi-task unlearnability for data protection. Essentially, current UEs are ineffective when deployed, and we seek to alert data protectors to these inherent limitations. Additionally, we aim to provoke significant attention, calling for a comprehensive reevaluation of how UEs handle cross/multi-task unlearnability.
- **For Future Work**: Providing an Effective Tool for Evaluating UE Methods. We provide a powerful tool to assess the practical unlearnability of UE methods, introducing two metrics (i.e., SAL and UD) during training. These metrics help data protectors and researchers predict UE effectiveness, minimizing resource waste. Moreover, SAL and UD could guide the development of more effective UEs by highlighting the essential characteristics they should possess.

### **5. Presentation Improvement**
- Modified Figure 1,2,4 to enhance their readability and clarity.
- Modified the Introduction and Section 3 (Sharpness-Aware Unlearnablity Explanation) with more detailed descriptions and more straightforward and insightful explanations.
- Added more details in the Appendix.

___

Best regards,

*The Authors*

---

### Meta-Review · Area_Chair_rtkx · 2024-12-11

**Metareview:**

In this work, the authors attempt to answer the question of "How Far Are We from True Unlearnability" from the perspective of model optimization. The authors observe the difference of convergence process between clean models and poisoned models on a simple model using the loss landscape and find that only a part of the critical parameter optimization paths show significant differences, implying a close relationship between the loss landscape and unlearnability. Consequently, the authors employ the loss landscape to explain the underlying reasons for UEs and propose Sharpness-Aware Learnability (SAL) for quantifying the unlearnability of parameters based on this explanation. Based on this, the authors propose an Unlearnable Distance (UD) metric to measure the unlearnability of data based on the SAL distribution of parameters in clean and poisoned models.

The novel aspect of this paper is that the authors aim to re-evaluate the true unlearnability of data in this paper and tackling this issue from the perspective of model optimization. Specifically, they introduced a metric called Sharpness-Aware Learnability (SAL) to quantify the unlearnability of parameters.

The finding of this paper is interesting, and the paper is also clearly written. The empircal analyses firmly validate the authors' finding.

**Additional Comments On Reviewer Discussion:**

All reviewers show positive scores to this paper, making this paper potentially publishable. As mentioned by the reviewers, the main finding of this paper is interesting. However, I hope the authors can still improve the quality of figures in preparing the final version if this paper is finally accepted.

---

### Decision · Program_Chairs · 2025-01-22

Accept (Poster)